# Effectiveness of virtual laboratory in engineering education: A meta-analysis

**Jiaxing Li**[1]*, **Wenhong Liang**[2]

**1** Philosophy Department, School of Humanities and Social Science, Xi'an Jiaotong University, Xi'an, China,
**2** School of Mechatronic Engineering, Xi'an Technological University, Xi'an, China

\* star630714@outlook.com

## Abstract

With the development of network and simulation technology, virtual laboratories have been widely popularized in engineering education. However, few studies have systematically analyzed and summarized the impact of virtual labs on the effectiveness of engineering education. This study aims to conduct a meta-analysis of published data on the impact of virtual laboratories on engineering students' performance. A total of 709 peer-reviewed publications on this topic were gathered from Web of Science and Scopus, and after strict inclusion criteria were applied, 46 studies from 22 publications were included in this meta-analysis. These studies were controlled experiments and pre-post designs with virtual labs as the intervention, reporting necessary descriptive summary statistics such as mean score comparisons and standard deviations of the two comparison groups. The results indicate that virtual laboratories are a significant predictor of engineering education outcomes, with an effect size (Hedges' g) of 0.686 (95% CI 0.414–0.959). Among these, the effect sizes for "learning motivation" and "learning engagement" are the highest across all types of results, at 3.571 (95% CI 3.042–4.099) and 2.888 (95% CI 2.419–3.357), respectively; this suggests that virtual labs are a key factor in motivating engineering students to engage in learning activities and pursue knowledge and skills. The results show that virtual labs currently lack the ability to completely replace hands-on labs in engineering education. However, they can inspire student motivation and engagement and compensate for the shortcomings of traditional lab facilities. Virtual labs have become an indispensable auxiliary tool in engineering experimental teaching. Therefore, consciously integrating virtual labs with physical experiences is a direction for sustainably developing engineering education in the future.

## 1. Introduction

Experimental education is indispensable in engineering education. Beyond theoretical concepts, laboratories play a crucial role in fostering essential engineering skills such as problem-solving, design proficiency, and effective troubleshooting. However, the integration of traditional laboratories in engineering education can be hindered by various factors such as equipment costs, time constraints, and infrastructure requirements. To mitigate these challenges, educators are actively seeking innovative technologies that enhance the inclusivity, creativity,

**Data Availability Statement:** The datasets that support the findings of this study are openly available in public database "figshare" with the DOI number 10.6084/m9.figshare.28014530. The link is: https://figshare.com/s/920eb123b889f46c50e3.

The datasets supporting the findings of this study are also available within the paper and its Supporting Information "Appendix 1. All data extracted from the publications that meet the inclusion criteria".

**Funding:** This work was supported by the Major Science and Technology Projects of the Autonomous Region: Integration and Demonstration of Key Technologies for Improving Quality, Increasing Yield, and Processing Comprehensive Utilization of Walnut (Item No.: 2021A02002-2). The funders had no role in study design, data collection and analysis, decision to publish, or preparation of the manuscript.

**Competing interests:** The authors have declared that no competing interests exist.

and efficacy of laboratory experiences. Nowadays, virtual laboratories have emerged as a prominent solution in engineering education, offering a popular and effective alternative to traditional lab setups.

The adoption of virtual laboratories in education can be traced back to the nascent stages of computer-based simulation and modeling. During the 1990s, initial virtual laboratories were predominantly employed as educational aids to complement conventional classroom teaching methods. For example, a 1994 study showed that using computer simulations from undergraduate engineering education before conducting physics experiments could significantly reduce students' time in the lab, decrease the number of requests for help from teaching assistants, and improve students' satisfaction with the lab experience [1]. With the advancement of technologies such as computer simulation, virtual reality (VR), augmented reality (AR), and interactive simulation, virtual laboratories have become more complex and interactive. They can create immersive, hands-on learning experiences for students. In recent years—especially during the COVID-19 pandemic—virtual laboratories have played a crucial role in providing remote learning support and uninterrupted continuing education [2]. As a result, virtual laboratories are widely used in the field of engineering education and have become a valuable tool for engineering educators. While offering the "actual" experience, whether in research or education, traditional laboratories (i.e., hands-on laboratories) are also known for their high costs associated with the required equipment, space, and maintenance staff [3]. Compared to traditional laboratories, simulation laboratories offer easier cost reduction, greater accessibility, time savings, a safe environment, and self-regulated learning [4]. Additionally, virtual labs foster students' learning of concepts and principles through simulations and representations of abstract phenomena. Finally, virtual labs are usually flexible and allow students to change the values of different variables and explore the experimental results faster than hands-on or remote experimentation [5].

One open issue related to virtual laboratories is associated with the specific semantics of the term. Definitions provided in the literature are inconsistent and confusing, and different terms are often used to define the same concept, such as remote, e-, web, online, and distributed learning laboratories. To avoid ambiguities in this paper, three major classification criteria are identified for the experiment at hand as follows: first, a virtual laboratory is an experimental environment constructed through computer simulation and virtual technology, where students conduct experiment operations and learn in a virtual environment [2]. In contrast, remote, e-, and web laboratories emphasize remote access and operation of actual experimental equipment [6]. Second, virtual laboratories are typically built based on computer software and simulation technology, aiming to provide a safe, controllable, and self-learning environment [5]. Remote laboratories connect experimental equipment via the network, allowing students to remotely access and operate the equipment for experiments [7]. Technical support includes network-communication, remote-control, and data-transfer technology. E-laboratories usually involve the use of electronic devices and software for experimental operations. Technical support includes electronic devices, circuit design software, and simulation software. Web laboratories provide experimental resources and environments through the network, allowing students to access the experiments online. Technical support includes network, server, and remote-access technology. The purpose of remote, e-, and web laboratories is to enable the remote access to and operation of experimental equipment, providing a more flexible and convenient experimental-learning environment [7]. Third, virtual laboratories help students understand experimental principles and skills by simulating experimental environments and processes [8]. Conversely, remote, e-, and web laboratories allow students to access and operate actual experimental equipment remotely to perform real experimental procedures [9]. However, in the field of engineering education, each of the above types of laboratories may

differ in implementation, technical details, and educational objectives. Yet, in practice, since these laboratory types all rely on computer, network, and remote-access technology to support students in conducting experiments and learning, they aim to provide an environment for student experimentation and learning and offer opportunities for independent student learning, enabling students to engage in experimental learning without time and location constraints. Therefore, they may have overlapping elements in specific experimental projects.

As an essential component of the STEM (Science, Technology, Engineering, and Mathematics) field, engineering education aims to cultivate students' knowledge and skills in engineering, enabling them to apply these abilities to solve problems and innovate designs. Engineering education not only covers the discipline of engineering itself but also encompasses knowledge in areas such as science, technology, and mathematics [10]. In recent years, the application scope of virtual labs in the field of engineering education has become increasingly wide, covering experimental operations and learning in various engineering disciplines and fields such as electrical engineering and automation, mechanical engineering, computer engineering, and civil engineering [11]. Currently, many studies have been conducted on the design and application of virtual laboratories in the field of engineering education. A considerable portion of these studies involves evaluating the actual role and effects of virtual laboratories in engineering education. This study aims to perform a meta-analysis of published data on the effects of virtual laboratories in engineering education to effectively evaluate the true role of virtual laboratories in the field of engineering education—especially the impact of virtual laboratories on engineering students' learning abilities, academic performance, thinking, and cognitive abilities.

## 2. Literature review

### 2.1 Simulation laboratories, remote laboratories, and virtual laboratories in engineering education

Simulation plays a crucial role in engineering education, especially in laboratory exercises. As early as 1928, Edwin Link developed a flight simulator called the "Link Trainer," considered to be the first simulation program used in the "Blue Box" [12] and engineering education. This simulator was used to train thousands of military pilots before and during World War II. Later, the applications of simulators were expanded to various industries such as aviation, chemicals, petroleum, nuclear energy, and other engineering applications. Finite Element Method (FEM) structural analysis tools and SPICE (a type of analysis software for circuit design) have completely transformed the simulation-software development process in engineering [13]. Engineering educators find simulation attractive owing to portability, ease of use, and cost effectiveness [14]. Simulation software, as a new teaching tool, is being applied in mechanical engineering education, playing an increasingly important role in teaching and learning; it can simulate the operation and behavior of real systems using mathematical models and computational algorithms, providing visual results and data analysis [15]. In engineering education, simulation software commonly used in teaching includes Altium, Proteus, MATLAB, CFX, Abques, and Multisim. The existing research has mainly discussed modeling methods based on various simulation software—for instance, Mohd and Makoto proposed a modeling and simulation approach using stereotypes and specializations of SYSML standards to facilitate mechatronics system design with the intention of filling the gap between system design and simulation in the context of mechatronics [16]. Other studies have analyzed the use of simulation software in mechanical engineering education from the perspective of technology characteristics. For example, Kenjo explained why simulation software is the special reference in engineering education from the vantage point of undergraduate education and the

retraining of technical instructors and working engineers [17]. Moreover, Ashkan presented an overview of current applications and the ongoing transition from physical experimentation to digital simulations and immersive simulated learning environments in engineering education; he also discussed that the Immersive Simulation-Based Learning (ISBL) approach provides a framework to reuse the models developed as part of their simulation projects for educational purposes [18]. However, the simulation environment has substantial limitations. The applicability and efficiency of the simulation depend on the software standards; thus, the actual knowledge and experience gained by students rely on the software's authenticity, constraints, and capabilities [19]. Moreover, the limited pre-designed inputs and outputs in the software restrict students' creativity [13].

The remote lab is a system that uses computer-based technology to connect students to the physical world, allowing them to access real devices in the lab through a web browser. With this system, students can send commands for preprocessing on their end, which are executed on the actual devices in the real lab through the server. The results of the experiments are then displayed to the students [13]. Switzerland's VITELS Virtual Internet and Telecommunications Laboratory project was the first to achieve the use of third-party hardware in remote network experiments [20]. Remote control has spread to the world's universities, particularly in engineering and science remote laboratories (RLs) [21]. The COVID-19 pandemic affected all areas of human activity; as a result, students did not receive face-to-face instruction, and access to the laboratory was limited or practically impossible. Especially in engineering education, students' practical abilities cannot be comprehensively developed [22]. If the required practical work is appropriately designed, remote-controlled experimentations allow students to execute it in a similar way as they would in physical laboratories [23]. Students can practice the experiments from any location at any time remotely through an Internet-connected computer client; therefore, during the COVID-19 period, remote labs provided a flexible, safe, and convenient experimental learning method for the field of engineering education. Currently, remote experiments have been applied to many engineering education disciplines, such as LabEAD, a remote lab project that aims to provide practical knowledge of learning opportunities for Brazilian engineering students [24]; the implementation of the digital twin (DT) concept for industrial equipment, which can partially solve the problem related to hardware unavailability by the remote learning process [25]; and remote operating-system design. In this regard, Dangfeng Pang and his research team built an online remote robotics experiment system using DT and Internet of Things (IoT) technology and adopted the ADDIE (Analysis, Design, Development, Implementation, and Evaluation) teaching method [22].

Usually, a virtual laboratory can be defined as an environment where experiments are performed or controlled through computer operations, simulations, or animations. It allows users to guide the experiment process and final product by using certain controllable experiment variables within the software. The experiments are typically graphic simulation models of real experiments. In other words, a virtual laboratory does not involve physical hardware, but it enables users to observe processes and final products through animations or simulations [23]. Simulations and modeling tools require huge computing resources and are limited to simplified models, ignoring real-world complexities [26]. However, with the development and advancement of technologies such as digital twins, AR, and VR, virtual labs combine the advantages of simulation environments and remote labs. Nowadays, virtual laboratories are often based on third-party virtual reality engines—for instance, game engines, which not only can be adapted to provide an educational virtual laboratory environment but also enable educational laboratory designers to flexibly assign laboratory tasks [27]. Compared with the remote "hands-on" laboratory, the virtual laboratory is cheaper and safer. The experimental results, which are calculated based on the mathematical models without model uncertainties

and practical disturbances, are closer to the theoretical results. In the past, the virtual laboratory could not be applied in experiments focused on data processing [28]. With the advancement of 3D visualization and VR technology, when web-based 3D technology is used as a supplement to web cameras in a remote-lab setting, it can eliminate the limitations of web cameras. First, it allows users to freely zoom in, zoom out, or rotate the 3D objects of the test platform; second, users can see all details about the test rigs; third, real-time simulation can be conducted when users are trying to monitor the experiment process [28]. These technologies are combined with virtual labs to form a new "hybrid lab." For instance, using AR offers an interactive framework that creates an augmented reality learning environment (ARLE) for the specific needs of electronics engineering laboratory hardware operations. It offers interactive 3D models of laboratory equipment, providing learners with preliminary training in equipment operation. This interface empowers users to control AR simulation seamlessly through the laboratory equipment [29].

## 2.2 Educational outcomes of virtual laboratories

Currently, the perspectives on the impact of virtual labs in the field of engineering education are two: one is about students' attitudes toward virtual labs, including their satisfaction, feedback after using virtual labs, and their experiences with simulation labs. This type of research typically involves designing appropriate user experience surveys or conducting interviews. Research results have shown that most students consider the experience of learning in virtual laboratories to be both easily comprehensible and enjoyable [30]. Most students are highly satisfied with their experience using virtual labs (these studies have often used the Likert scale) [31]. Cognitive partnerships are formed between students and the virtual laboratory artifact, leading to a rich learning experience [32]. The virtual experiment system provides students with a rich, efficient, and expansive experimental experience—in particular, the flexibility, repeatability, and visual appeal of a virtual platform could promote the development of students' abilities in active learning, reflective thinking, and creativity [33]. Further, the virtual laboratory is effective in enhancing curiosity, learning motivation, and engagement among students [34]. Another research perspective has focused on students' performance and pedagogical achievement; evaluation studies in certain STEM contexts have shown that students' achievement or performance in such virtual labs is consistent with that found in traditional face-to-face laboratory experiences [35]. Virtual laboratories offer learners important pedagogical value, which results in a substantial positive impact on operational skills in engineering education [29]. Additionally, the virtual simulation experiment has a stronger teaching effect than the field experiment [36]. Students who used the AR application presented a better academic performance than those who participated in the traditional laboratory [37]—to be more specific, students in the virtual lab condition acquired better conceptual understanding and also developed better procedural skills than students in the traditional condition [38]. This type of research often adopts controlled experiments or pre-post designs, using virtual laboratories as an intervention in engineering education, setting up experimental and control groups and then comparing the differences in results between the two groups, or measuring participants' specific outcomes after receiving the intervention and then comparing the results before and after intervention, ultimately determining the effectiveness and impact of virtual laboratories in engineering education. This meta-analysis aims to collect such publications, integrate the results of multiple independent studies, and quantify and evaluate overall effect sizes to obtain a more comprehensive and objective conclusion about the effectiveness of virtual labs in engineering education.

Overall, with the advancement of technology and the emphasis on remote and digital learning, virtual laboratories have become an important component of engineering education. They

offer flexibility, accessibility, and the ability to simulate complex experiments that are difficult or costly to conduct in physical laboratories. The implementation of virtual laboratories varies across different institutions and engineering disciplines, and the effectiveness of these implementations may differ depending on the technology used, integration with the curriculum, and specific learning objectives. Meta-analysis can synthesize these diverse findings to provide a clearer picture of overall effectiveness. Additionally, while individual studies may explore specific aspects of virtual laboratories, they may not provide a comprehensive analysis of overall effectiveness across different contexts and variables. Meta-analysis can fill this gap by aggregating data from multiple studies, thereby identifying common trends and insights.

## 3. Methodology

Meta-analysis is a research method for systematically combining and synthesizing findings from multiple quantitative studies in a research domain [39]. This study chose the meta-analysis methodology for several reasons. First, a considerable portion of research on the design and utilization of virtual laboratories for engineering education practices used two sets of research designs (experimental, quasi-experimental, or comparative) to evaluate and assess virtual laboratories. For example, Yin Xiaoju, in his study of a virtual simulation platform for the teaching exploration of a practice course on wind-turbine assembly, developed and discussed the teaching of a simulation working process based on the wind-turbine assembly process and performance characteristics, and the results and questionnaires of students in three classes were counted [40]. Furthermore, many studies have also adopted experimental, quasi-experimental, or comparative methods aimed at quantifying the effectiveness of virtual laboratories in engineering education specifically, such as Faridaddin Vahdatikhaki, who conducted a quasi-experimental study that divided 92 students into control and experimental groups using cluster sampling. The control group received traditional lab training, while the experimental group engaged with the virtual-laboratory training environment [41]. A systematic review of existing studies can not only help us focus on the effect size across these studies but also provide a more accurate summary and description of the impact of virtual laboratories on engineering students' performance in professional learning. Second, by extracting and analyzing all available published data from a systematic review (i.e., quantifying and combining the results of individual studies) and then calculating the effect size of each study, we can compute a summary effect size from these studies [42]. Finally, meta-analysis will provide us with a more quantitative and statistical conclusion, taking into consideration the strength of the effect size in each empirical study as well as the synthesis of these studies and a more precise perspective [43].

In designing the methods for this study, we referred to the review protocol developed by Andy P Siddaway, which specifies instructions for conducting meta-syntheses [44]. Our review proceeded through the following stages: (i) define inclusion/exclusion [43] and database searching; (ii) from abstract reading to full-text sifting to screen scope; (iii) map the literature through a scoping review; and (iv) conduct in-depth analysis for the meta-synthesis review [45]. During this meta-analysis, we followed the PRISMA 2020 statement [46], which outlines widely accepted guidelines for systematic reviews.

### 3.1 Define inclusion and exclusion

Before reviewing the literature, clear criteria were set to determine which research papers from the bibliometric database would be included or excluded. These criteria were carefully defined to ensure the relevance and quality of the selected full-text publications.

To ensure the effectiveness of publication retrieval and collection, we developed our search strategy based on the PICO framework—a structured tool used to formulate searchable clinical or research questions, widely applied in fields such as evidence-based medicine, clinical research, and public health. The four letters of PICO represent the following elements:

P: Patient, population, predicament, or problem.

I: Intervention, exposure, test, or other agent.

C: Comparison intervention, exposure, tests, and so on, if relevant.

O: Outcomes of clinical importance, including time, when relevant [47].

The specific steps of PICO format are (1) identifying the Population (patients) by considering how it was recruited and whether an appropriate target population was identified; (2) determining the Intervention, exposure, or test that to which the population was subjected; (3) defining the Comparison (or Control) group and how the participants were selected or allocated; and (4) establishing whether clinically important Outcomes were measured in a blind and/or objective fashion and at an appropriate time from a clinical perspective [47].

Regarding our research theme, the four elements of PICO can be confirmed as

Patient/Population/Problem: The effectiveness of engineering education.

Intervention: Virtual laboratory.

Comparison: Engineering education without the use of a virtual laboratory.

Outcomes: The academic performance of engineering students and quality of instruction in professional courses.

The publications we need in this study must meet the following criteria: first, they should involve assessing engineering education effectiveness; second, they should conduct empirical research using virtual laboratories as an intervention method, employing experimental methods such as control trials or pre-test and post-test designs; third, the control group should consist of engineering students who have not received virtual laboratory education, with participants randomly assigned to either the experimental or control group; fourth, the outcomes must be measured in objective fashion, with assessment standards that can objectively reflect students' academic performance and the quality of instruction in professional courses in engineering education, including academic achievement, classroom participation, homework completion, and student engagement.

Therefore, in this study, the search terms and inclusion criteria were as follows: first, "virtual laboratory" (as an intervention method) was the core keyword. Since virtual laboratory is the full name, and "Virtual Lab" and "VLab" are both abbreviations of virtual laboratory, our search strategy included a keyword search of "Virtual Lab and "VLab." Second, key indicators of learning effectiveness include not only effects and effectiveness but also student performance, academic achievement, classroom participation, homework completion, and student engagement; therefore, when conducting searches, we combined these topic words. Third, to ensure the credibility and effectiveness of this meta-analysis, we selected empirical studies over purely theoretical research, specifically studies utilizing controlled experiments as their methodology. Key terms related to empirical research (e.g., Empirical, Observational, Experimental, Survey) and terms related to controlled experiment methods (e.g., Control Experiment, Experimental Group) appeared in the abstracts of studies that are empirical and employ controlled experiment methods; therefore, we imposed restrictions on abstract-related keywords when searching.

We selected Web of Science and Scopus as two authoritative databases to search for literature relevant to the topic. The keywords used in this research include "virtual laboratory," "Virtual Lab," "VLab," and "engineering," as well as "education," "learning," "study," "class," and "course," which link virtual laboratory with engineering education. Additionally, keywords include "effects," "effectiveness," "impact," "academic achievements," "completion of assignments," "knowledge," "skills," "abilities," "participation," "interaction," "motivation," "interests," "attitudes," "feedback," "evaluation," which indicate the possible impact of virtual laboratory on engineering education. In addition, the retrieval strategy also includes keywords in the abstract such as "Empirical," "Observational," "Experimental," "Survey," "Data," "Research Design," "Statistical Analysis," "Sampling," "Control Experiment," "Experimental Group," "Control Group," "Random Allocation," "Intervention Control," "Controlled Variables," or "Experimental Design," "Causal Relationship," "Reliability," "Effect Evaluation" (these terms indicate that the study adopts empirical research methods with two-group research designs). The search date was June 24, 2024.

After completing data collection, we started screening studies to include in this meta-analysis. Therefore, we established strict inclusion criteria to determine whether preliminary studies qualified for inclusion in this systematic review. Studies meeting the following criteria were included in the meta-analysis: (i) using a two-group research design focusing on comparing the impact of various forms of virtual laboratory with traditional laboratory on engineering student performance; (ii) involving engineering education at various stages; and (iii) reporting necessary descriptive summary statistics, such as the mean achievement scores and standard deviations of the two comparison groups.

## 3.2 Database searching and data collecting

When we started this research, we noted the absence of published summaries of this theme, and did not encounter any engineering education researchers who had methodically delved into this domain. For this study, the data-collection procedure comprised rigorously and systematically locating all relevant research related to virtual-lab interventions on the performance of students in engineering education. We searched two general databases, Web of Science and Scopus.

The Web of Science database covers literature and journals in multiple subject areas, regularly updating its content to maintain the timeliness and accuracy of information; thus, it met the requirements for database searching in this research. For this meta-analysis, we searched Web of Science using search terms, search queries, and exclusion criteria, with a combination of "AND," "OR," and "NOT" Boolean search operations to ensure that the retrievals or outputs were precise. According to our search strategy, our final search string was as follows: TS = (virtual laboratory OR Virtual Lab OR VLab) AND TS = (engineering) AND TS = (education OR learning OR study OR class OR course) AND TS = (effects OR effectiveness OR Impact OR academic achievements OR completion of assignments OR knowledge OR skills OR abilities OR participation OR interaction OR motivation OR interests OR attitudes OR feedback OR evaluation) AND AB = (Empirical OR Observational OR Experimental OR Survey OR Data OR Research Design OR Statistical Analysis OR Sampling) AND AB = (Control Experiment OR Experimental Group OR Control Group OR Random Allocation OR Intervention Control OR Controlled Variables OR Experimental Design OR Causal Relationship OR Reliability OR Effect Evaluation). Based on this, we excluded review articles and abstracts from the types of studies that could be included, and limited the research fields to Engineering, Computer Science, Education, Educational Research, Automation Control Systems, Instruments Instrumentation, Arts Humanities Other Topics, Psychology, Behavioral Sciences, Social Sciences Other

Topics, Social Issues, or Sociology. However, we did not apply any additional limitations, such as year of publication or language. In the end, we obtained 695 publications from the Web of Science database.

Scopus was selected as a bibliometric resource owing to its comprehensive coverage of peer-reviewed educational literature, encompassing a wider range of academic journals across the fields of science, technology, medicine, natural sciences, and social sciences. The search strategy was similar to that of Web of Science, performed using a combination of search terms, search queries, and exclusion criteria with a combination of "AND," "OR," and "NOT" Boolean search operations. However, the difference was that the search system of the Scopus database can directly lock in the studies to be included in this meta-analysis by searching for the publication date, title, abstract, document title, document type, journal name, and main keywords in the reference field. The final search string was as follows: ABS ("Empirical" OR "Observational" OR "Experimental" OR "Survey" OR "Data" OR "Research Design" OR "Statistical Analysis" OR "Sampling") AND ABS ("Control Experiment" OR "Experimental Group" OR "Control Group" OR "Random Allocation" OR "Intervention Control" OR "Controlled Variables" OR "Experimental Design" OR "Causal Relationship" OR "Reliability" OR "Effect Evaluation") AND TITLE-ABS-KEY ("virtual laboratory" OR "Virtual Lab" OR "VLab") AND TITLE-ABS-KEY ("engineering") AND TITLE-ABS-KEY ("education" OR "learning" OR "study" OR "class" OR "course") AND TITLE-ABS-KEY ("effects" OR "effectiveness" OR "Impact" OR "academic achievements" OR "completion of assignments" OR "knowledge" OR "skills" OR "abilities" OR "participation" OR "interaction" OR "motivation" OR "interests" OR "attitudes" OR "feedback" OR "evaluation"). In the end, we obtained 45 publications from the Scopus database.

Finally, we obtained 740 records in this search, which were added to a shared literature management system in Endnote X8. After manually excluding 31 duplicate records through browsing, we retained 709 unique records (Fig 1).

## 3.3 Abstract reading and full-text sifting

We developed two detailed codebooks using Microsoft Excel to explain the steps and criteria for abstract screening and full-text sifting (Table 1). To ensure the reliability of information screening and statistical results, we used the reference management software EndNote X8 to manage and document the literature-screening process. EndNote X8 is a powerful reference management software designed to help researchers manage references and generate citations. Users can easily import, organize, and manage literature, create different folders for classification and retrieval, and share their reference libraries with others, facilitating teamwork. The number of publications excluded from each step is shown in Fig 1.

After these steps, publications that were irrelevant to the theme and those based solely on students' subjective feedback (such as studies assessing the perceived usability and user satisfaction of various forms of virtual laboratories) were excluded. Ultimately, publications that aligned with the theme and were based on objective evaluation criteria (such as skill tests and exams) were included. These studies employed a two-group research design to compare the effects of various forms of virtual laboratories with traditional laboratories on engineering students. Additionally, these studies reported necessary descriptive summary statistics.

During the abstract-reading phase, the division of work was as follows: one person was responsible for the initial screening of abstracts from the literature database to determine whether they met the inclusion criteria.; another person reviewed and verified the first person's screening results to ensure consistency and accuracy. In cases of uncertain abstracts, a discussion was needed to reach a decision, ensuring that all relevant literature was evaluated. Finally,

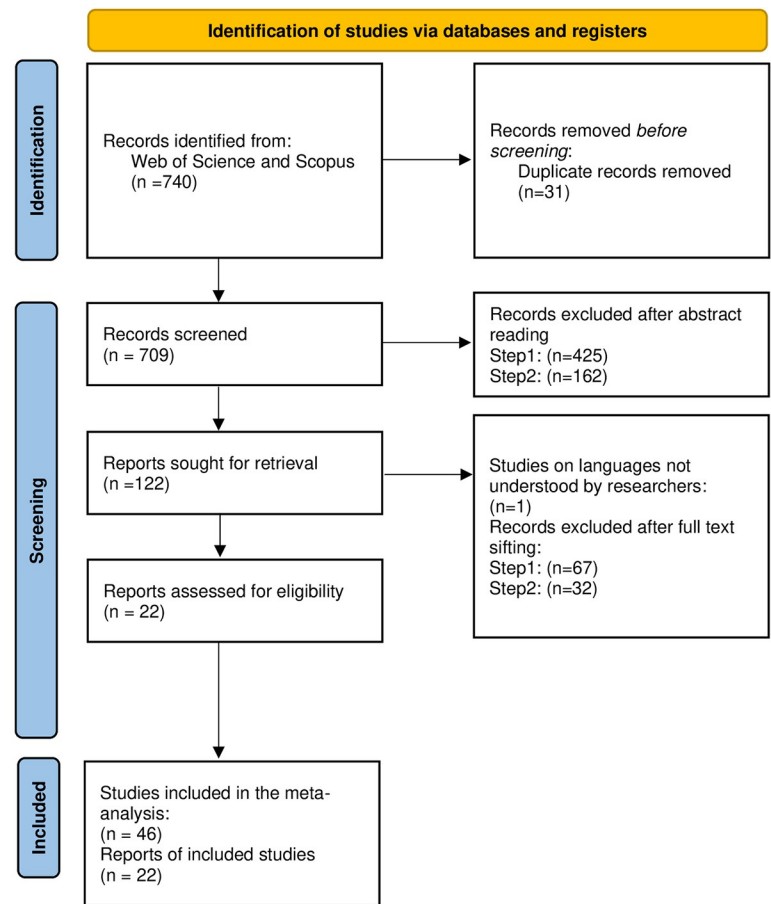

*Consider, if feasible to do so, reporting the number of records identified from each database or register searched (rather than the total number across all databases/registers).

**If automation tools were used, indicate how many records were excluded by a human and how many were excluded by automation tools.

*From:* Page MJ, McKenzie JE, Bossuyt PM, Boutron I, Hoffmann TC, Mulrow CD, et al. The PRISMA 2020 statement: an updated guideline for reporting systematic reviews. BMJ 2021;372:n71. doi: 10.1136/bmj.n71

**Fig 1. Flow diagram of the selection process.**

we eliminated 425 publications that were unrelated to the topic by reading the titles and abstracts. We also excluded 162 publications that only introduced the design and development of virtual laboratories and the application of virtualization technology, reducing the number of publications needing further screening to 122.

In practical engineering education, virtual laboratories often overlap, to some extent, with remote, e-, web, online, and distributed learning laboratories. The core of a virtual laboratory differs from that of other types of laboratories, in that it provides students with a safe, controllable, autonomous virtual experiment and learning environment, rather than assisting them in conducting real experiments through remote access and operation of laboratory equipment.

**Table 1. Steps and criteria for abstract screening and full-text sifting.**

| | | |
|---|---|---|
| Abstract screening | Step 1: Identify the research topic of the publications. | Include: Studies on methods, technologies, and developments related to engineering education, or studies involving participants and settings in engineering education (such as students, instructors, administrators). |
| | | Exclude: Studies unrelated to engineering education. |
| | | Include: The topic of the study is about or involves the effectiveness of virtual laboratories. |
| | | Exclude: Issues not related to the virtual laboratory. |
| | | Exclude: Focused on the design and development of virtual laboratories and the application of virtualization technology. |
| | Step 2: Identify the research type of the publications. | Include: Empirical research conducted using two-group research design. |
| | | Exclude: Reflective practice, popular science, and nonscientific publications |
| Full-text sifting | Step 1: Identify the research topic of the publications. | Include: The topic of the study is indeed about virtual laboratories, not about remote, e-, web, online, and distributed learning laboratories and their effectiveness in engineering education. |
| | | Exclude: The topic of the study is not virtual laboratories, or the topic of the study is unclear. |
| | Step 2: Identify the publication has utilized reliable and effective experimental methods. | Include: The design of the experimental group is clearly described; the treatment, intervention, discipline, and sample size, as well as the means (M) and standard deviation (SD) of the control group, are specified. |
| | | Exclude: Insufficient information provided. |

Note: Publications excluded in an earlier step were not screened for the remaining steps.

Therefore, determining whether the topic involved creating experimental environments using computer simulation and virtual technology to simulate experiments and processes was essential. In full-text sifting, we focused on filtering publications about virtual laboratories but not about remote, e-, web, online, and distributed learning laboratories. After establishing that the research topic met our inclusion criteria, the next step was to determine whether the publication had adopted reliable and effective experimental methods, including whether the design of the experimental group was clearly described and the treatment, intervention, discipline and sample size, as well as the means (M) and standard deviation (SD) of the control group, were specified. During the full-text sifting phase, we had to determine whether the theme of the publication was indeed the effectiveness of virtual laboratories in engineering education and confirmed if the study provided sufficient data to support its conclusions. The division of work during the full text reading phase was as follows: one person was responsible for obtaining the full text of all articles that passed the abstract screening, determining whether they met the inclusion criteria. If they did, key information and data were extracted (S1 Appendix. All data extracted from the publications that meet the inclusion criteria). Publications that did not meet the inclusion criteria were also recorded, along with an explanation for their exclusion (S2 Appendix. Publications identified in the literature search). Another person was responsible for reviewing and verifying the accuracy of the data and information extracted from all the publications included in this research, as well as for examining the excluded articles and discussing any discrepancies in the review results until a consensus was reached. Finally, after the participants' review and communication, we narrowed down the scope of the publications to 22 articles, with 46 separate studies.

**Table 2. Example of coding schema.**

| | authors and publication year | research country/region | level of education | research setting | research design | sample size | | intervention | outcomes | EG | | CG | |
|---|---|---|---|---|---|---|---|---|---|---|---|---|---|
| | | | | | | EG | CG | | | M | SD | M | SD |
| 1 | (Wang et al., 2024) | China | Tertiary Education | Industrial Robotics | Controlled Experiments | 150 | 90 | MSC-vslab | Academic achievement | 85.040 | 8.810 | 82.620 | 8.810 |
| | | | | | | | | | | 85.760 | 6.770 | 81.670 | 7.930 |

Note: EG = Experimental Group; CG = Control Group; M = Means; SD = Standard Deviation

## 3.4 Coding of moderator

After individually examining the selected publications to ensure that they met the inclusion criteria, we developed a systematic and reliable coding procedure for extracting data from the targeted studies. The objective of the formalized coding process was to capture all key information that could then be formatted for easy extraction of data. As shown in Table 2 (an example of relevant variables), the coding scheme included the following variables: title, authors, publication year, research country, level of education, research setting, research design (including controlled experiment and Pre-Post designs), type of intervention, sample size, outcome type (including academic achievement, operational skill, cognitive ability, etc.), level of education (including primary, secondary, tertiary, and vocational), and the means (M) and standard deviation (SD) of the experimental and control groups.

## 3.5 Subgroup and moderator analysis

To compare the effectiveness of virtual labs on different types of outcomes in engineering education, we added subgroups. We summarized and analyzed the outcomes included in publications for this meta-analysis, establishing subgroups of learning outcomes in engineering education, including Academic Achievement, Operational Skill, Cognitive Ability, Knowledge Acquisition, Cognitive Load, Learning Motivation, and Learning Engagement. These subgroups might have impacted the results of the meta-analysis.

Additionally, we conducted two separate moderator analyses to explore potential variations due to other present factors. The two potential moderators considered were (i) country/region and (ii) publication year.

## 4. Results

Based on the extracted data, we conducted statistical analyses and calculations on the heterogeneity, effect sizes, publication bias, and sensitivity of the included publications. To ensure the reliability and standardization of the results, we used the Comprehensive Meta Analysis software for data calculations and figure creation.

### 4.1 Computation of heterogeneity

According to the methods proposed by Higgins and Thompson [48] in their research for assessing heterogeneity and interpreted as the percentage of between-study variance in meta-analysis, namely the Q statistic and $I^2$ statistic, we calculated the heterogeneity of the publications included in this study, as shown in Table 3.

Based on the Q-test, the test for heterogeneity was significant, indicating high variation among the studies ($p < 0.0001$; $I^2 = 94.768\%$). High heterogeneity is particularly common in educational research studies, and a possible explanation for this finding may be due to a combination of variability as well as small effect sizes found in some of the publications [42].

**Table 3. Effect and heterogeneity.**

| Model | Number Studies | Effect size and 95% confidence interval | | | Heterogeneity | | |
|---|---|---|---|---|---|---|---|
| | | Variance | Lower limit | Upper limit | Q-value | I-squared | P-value |
| Fixed | 46 | 0.001 | 0.479 | 0.598 | 860.097 | 94.768 | <0.0001 |
| Random | | 0.019 | 0.414 | 0.959 | | | |

Therefore, we adopted a random effects model to estimate the true effect sizes and weights of each study to derive the summary effect size.

## 4.2 Computation of summary effect sizes

This study used Hedges' g formula to calculate the effect size of the mean difference in a controlled experiment, measuring the magnitude of the mean difference effect between two groups. Compared to Cohen's d, Hedges' g addresses bias issues in small sample sizes, making it more precise when sample sizes are small in this study. The calculation formula for Hedges' g is as follows:

$$G = \frac{M_1 - M_2}{SD_P}$$

In which $M_1$ and $M_2$ are the means of two groups, and $SD_P$ is an estimate of the combined standard deviation of the two groups. Hedges' g can be used to compare the effect size of mean differences between two independent samples. Typically, effect sizes around 0.2 indicate a small effect, 0.5 a moderate effect, and 0.8 or higher a large effect.

As shown in the forest plot in Fig 2, the effect size (Hedges' g) was 0.686 (95% CI 0.414–0.959), providing evidence that the virtual laboratory is a significant predictor of engineering education outcomes. Each study in the forest plot is represented by a black square (indicating its estimated effect size) and its area proportional to study weight, with the horizontal line indicating the 95% CI, and the corresponding black diamond representing the 95% CI of 46 studies. Owing to the involvement of multiple independent studies in some publications, the following distinctions were made in all forest plots: a suffix of "A" indicates independent studies focused on students' Academic achievement; "O" indicates independent studies focused on students' Operational skill; "C" indicates independent studies on Cognitive ability; "K" indicates independent studies focused on students' Knowledge acquisition; "CL" indicates independent studies on students' Cognitive Load; "L" indicates independent studies focused on students' Learning motivation; and "E" indicates independent studies focused on students' learning Engagement. If the same publication involved multiple similar studies, we added Arabic numerals after the uppercase letter suffix for differentiation.

## 4.3 Tests for publication bias

To investigate publication bias, we examined the symmetry of funnel plots. As shown in Fig 3, a few study results deviate from the center line of the funnel plot, and the bottom of the funnel plot shows missing data points. This indicates a certain degree of publication bias, where small sample studies or negative results may not have been published, leading to a bias in the overall effect size.

## 4.4 Sensitivity analysis

We performed sensitivity analysis to evaluate whether the pooled effect size was influenced by individual studies, that is, to assess the influence of the individual studies on the overall results

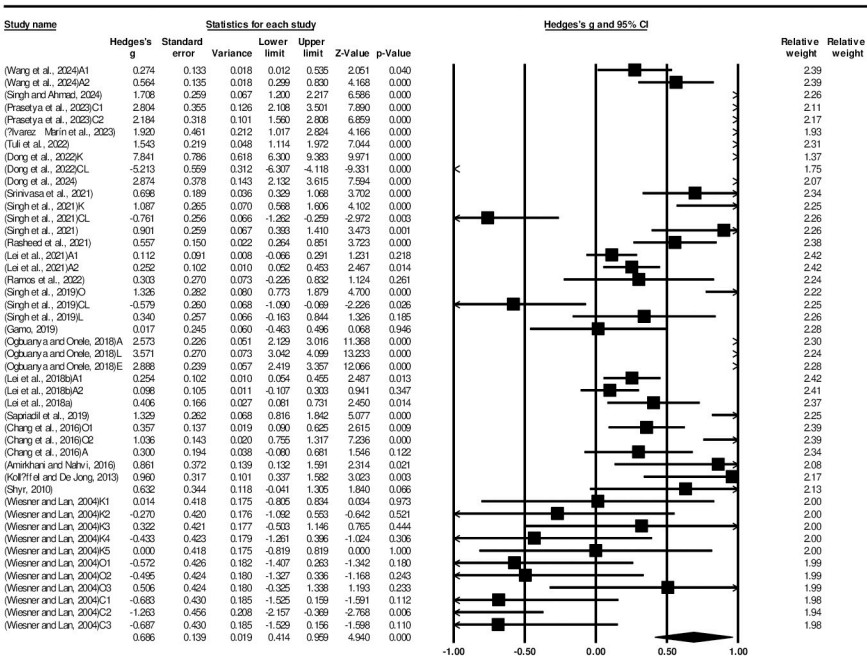

Note: Academic achievement =A; Operational skill=O; Cognitive ability=C; Knowledge acquisition=K; Cognitive load=CL; Learning motivation=L; Learning Engagement=E;

**Fig 2. Forest plot and summary effect.**

by omitting one study at a time. For this sensitivity analysis, a summary of the effect sizes was compared together with their 95% Cis, and the results confirmed the robustness to all ($n = 46$) one-by-one study removals. The width of the confidence intervals varies slightly when different studies are excluded, but overall, they did not cross zero, which indicates that the results are statistically significant. All analyzed Z values are greater than 1.96, and the p values are less

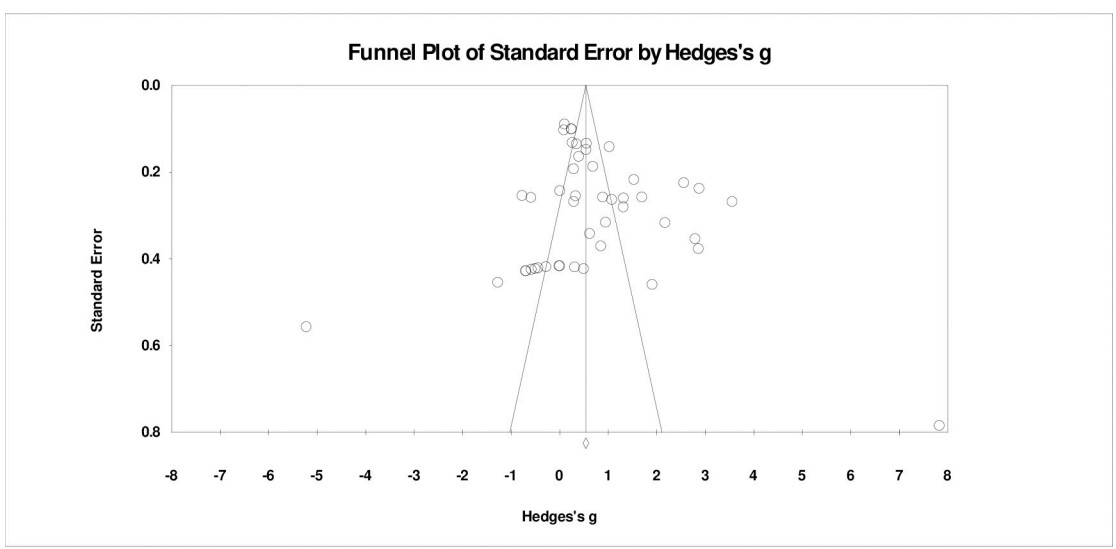

**Fig 3. Funnel plot.**

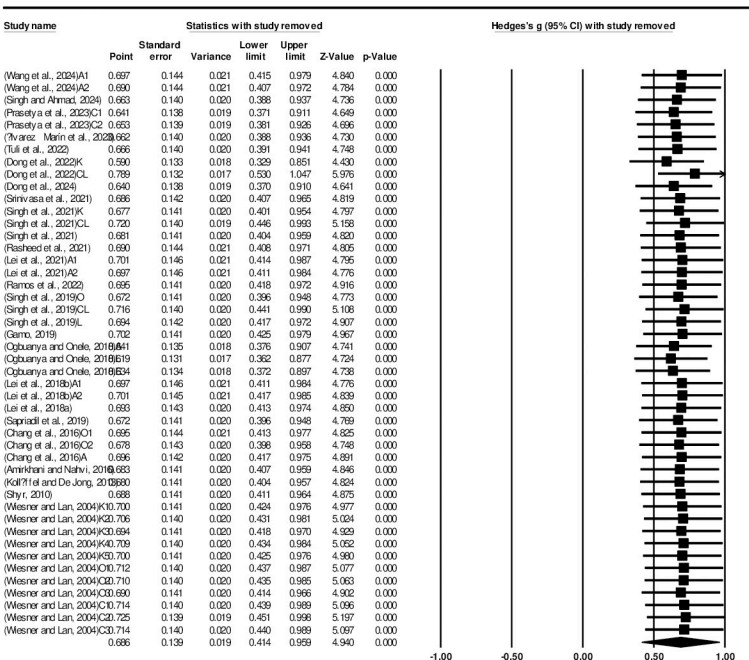

Note: Academic achievement =A; Operational skill=O; Cognitive ability=C; Knowledge acquisition=K; Cognitive load=CL; Learning motivation=L; Learning Engagement=E;

**Fig 4. Forest plot for sensitivity analysis.**

than 0.0001, indicating that regardless of which study is excluded, the results are statistically significant (Fig 4). The overall effect size estimate (Hedges' g) varies between approximately 0.590 and 0.789 when different studies are excluded, and the change magnitude is approximately 0.2. Thus, overall, the intervention has a positive and significant impact on the outcomes, and the change in effect size is relatively mild, with no significant fluctuations. That is to say, despite the methodological differences among various studies, the results demonstrate good robustness.

## 4.5 Computation of subgroup effect sizes

To compare the effects of intervention on different types of outcomes, we categorized the learning outcomes of 46 studies into Academic Achievement, which refers to the impact of virtual labs on students' exam scores (usually midterms or finals, standardized tests); Operational Skill, which refers to the impact of virtual labs on students' ability to conduct experiments; Cognitive Ability, including students' critical and creative thinking abilities; Knowledge Acquisition, which here refers to the knowledge students acquire in the virtual lab during the lesson, tested immediately after the intervention (non-standardized test results); Cognitive Load, the only category with a negative effect direction; Learning Motivation, also known as Learning Interest; and Learning Engagement, representing students' level of behavioral participation in this study. The results are shown in Fig 5.

The results indicate significant differences in the effect sizes of different types of outcomes in engineering education when using virtual labs. Among them, the effect sizes of "Learning motivation" are the highest among all types of outcomes; Hedges' g is 3.571, (95% CI 3.042–

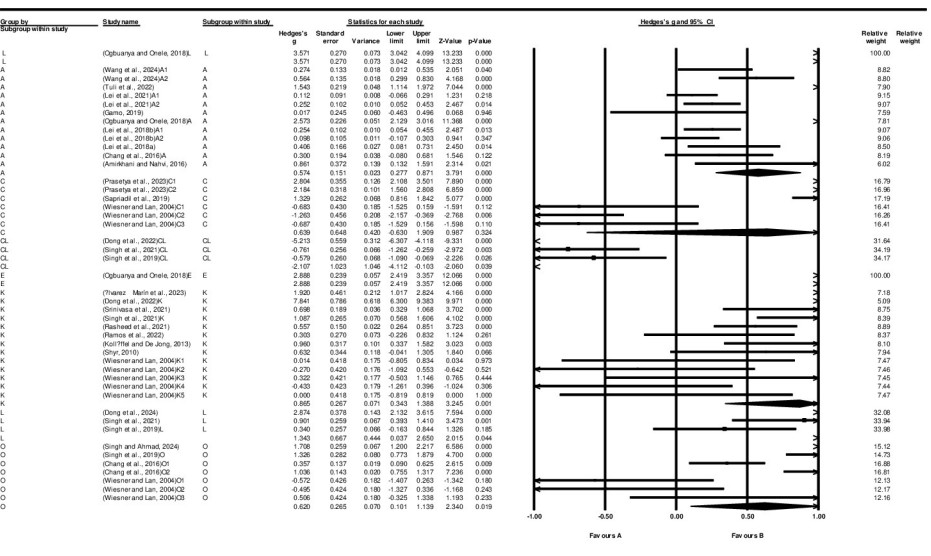

**Meta Analysis**

Note: Academic achievement =A; Operational skill=O; Cognitive ability=C; Knowledge acquisition=K; Cognitive load=CL; Learning motivation=L; Learning Engagement=E;

**Fig 5. Forest plot and subgroup effect.**

4.099), which indicates that virtual labs are a crucial factor in motivating engineering students to engage in learning activities and pursue knowledge and skills. Second only to "Learning motivation" is "Learning Engagement," with an effect size of 2.888 (95% CI 2.419–3.357), which indicates that the impact of virtual labs on student behavioral participation is also significant. "Knowledge acquisition," "cognitive ability," "operational skill," and "academic achievement" have effect sizes of 0.865, 0.639, 0.620, and 0.574 respectively. Cognitive Load, the only category with a negative effect direction, has an effect size of -2.107 (95% CI -4.112–-0.103).

### 4.6 Computation of moderator effect sizes

The two potential moderators considered were (i) country/region; and (ii) publication year (Figs 6 and 7, respectively).

The results show that Nigeria's publications had the highest effect size, at 2.995 (95% CI 2.437–3.553), while the USA's publications had the lowest effect size at 0.006 (95% CI -0.318–0.330). Additionally, the variability of the publication year effect sizes is as follows: the highest effect size was from a study in 2023, at 2.340 (95% CI 1.845–2.834); and the lowest effect size was from a study in 2004, at -0.311 (95% CI -0.606–-0.017). Overall, the variability of the effect sizes by publication year shows a general increasing trend over the years.

## 5. Discussion

This meta-analysis aimed to provide a quantitative and systematic review of experimental studies, attempting to summarize the research on the impact of virtual laboratories on

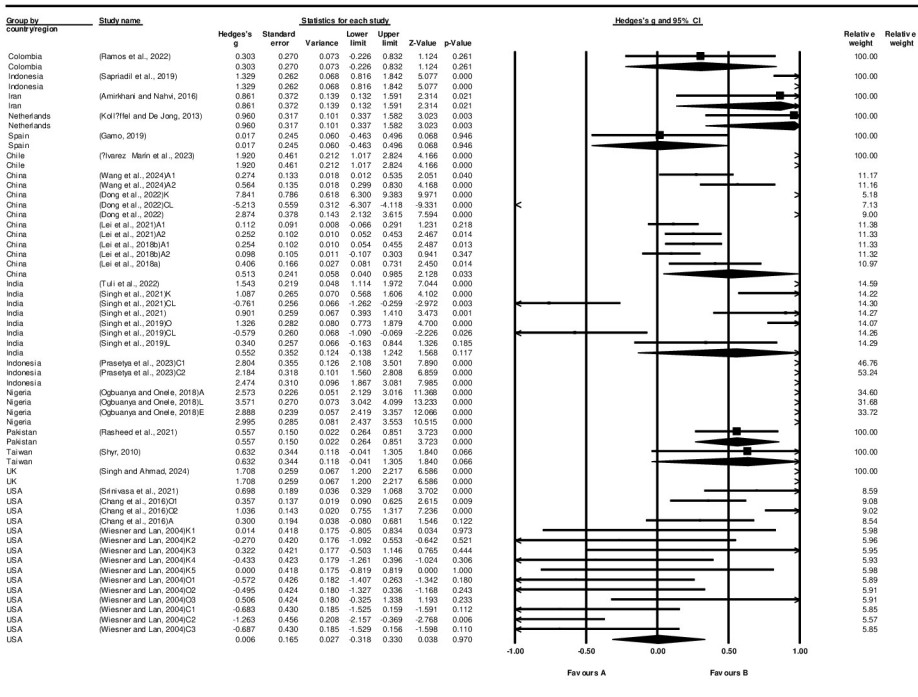

Note: Academic achievement =A; Operational skill=O; Cognitive ability=C; Knowledge acquisition=K; Cognitive load=CL; Learning motivation=L; Learning Engagement=E;

**Fig 6. Forest plot and country(region).**

engineering education. Many studies have shown the effectiveness of virtual laboratories, recognizing that in engineering education, virtual labs are a feasible alternative to traditional labs. With the advancement of supporting technologies such as AR, VR, and simulation, virtual labs are developing more features. Therefore, analyzing the impact of virtual labs on engineering education and understanding the factors that drive these effects is essential. The effect size found in this study reflects a significant positive impact of virtual labs on engineering education. By analyzing the effect sizes of subgroups, we found that the virtual laboratory in engineering teaching practice mainly influences students in the motivational aspect, including by enhancing students' willingness to actively participate in learning [49], sparking their interest in learning [27], and encouraging them to invest more effort in their professional studies. In terms of student performance and pedagogical achievement, the utility of virtual labs for students in acquiring knowledge and enhancing skills during class and practical activities is greater than their utility in improving exam scores. This tendency has been demonstrated in studies with a pre-post design. Measurements or assessments were conducted on students before they used virtual labs to establish baseline data prior to the intervention, determining the status of the subjects being measured before the start of the research. After virtual labs were used as an intervention, the students' situation was measured or assessed; this revealed that the same participants showed higher cognitive and operational abilities [27, 50] as well as lower cognitive load after undergoing virtual-lab interventions. Therefore, virtual labs in the field of engineering education are an effective learning tool that combines engineering

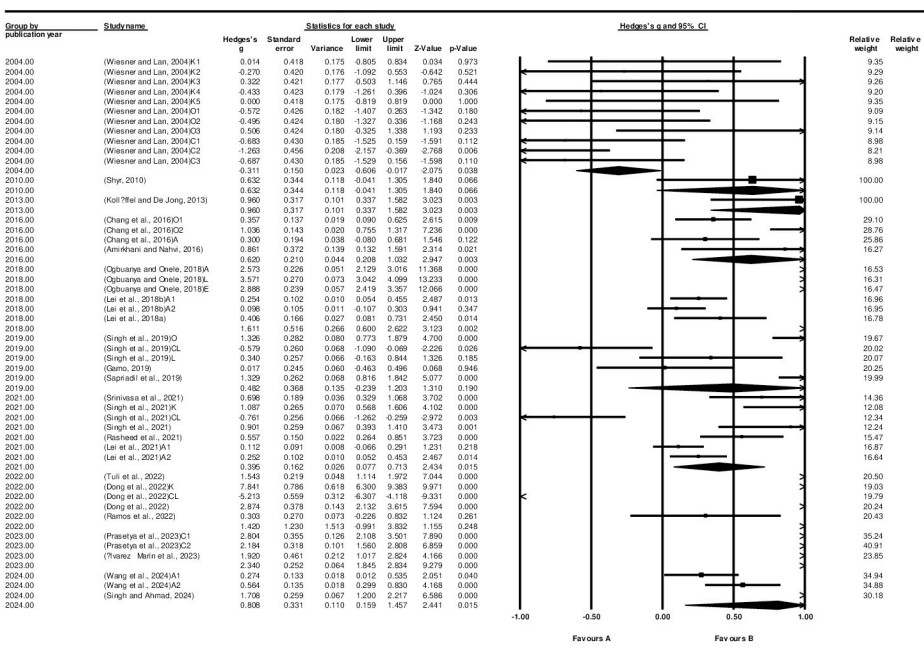

**Meta Analysis**

Note: Academic achievement =A; Operational skill=O; Cognitive ability=C; Knowledge acquisition=K; Cognitive load=CL; Learning motivation=L; Learning Engagement=E;

**Fig 7. Forest plot and publication year.**

education with practical needs [51]; it assists students in understanding abstract concepts and principles [38, 52, 53], enhances learning outcomes in terms of cognitive abilities [50, 54], and fosters enriched learning interactions [29], or it aims to train engineering students about the laboratory hardware [55, 56] and ease their difficulties while they operate laboratory equipment [57]. In summary, virtual laboratories have not yet completely replaced physical laboratories in the field of engineering education. However, they can compensate for the shortcomings of traditional laboratory facilities by minimizing the need for specialized laboratory infrastructure while maintaining the valuable essence of hands-on experience, thereby improving the accessibility of engineering education [35]. Moreover, the intentional blending of virtual labs with physical experiences can be especially effective in this regard [58].

Based on the analysis of moderator effect sizes, we found certain differences in effect sizes among different countries/regions and publication years. Publications from Nigeria showed a significant effect size (2.995), while publications from the United States indicated an extremely low effect size (0.006). This suggests that stronger factors or contextual conditions in Nigeria may make the relevant effects more pronounced; therefore, the differences in effect sizes in our research may be due to the varying developments in social, economic, or cultural contexts across different countries. However, owing to the limited number of studies from different countries, the analysis of the moderating effects in this research can only serve as a hypothesis and reference. We hope that future research will provide more reliable evidence on this issue.

Additionally, the effect sizes by publication year showed a higher effect size (2.340) for the recent study from 2023 and a negative effect size (-0.311) for the study from 2004. This

phenomenon often suggests that the latest research provides stronger support and evidence for the phenomenon of interest, which may be related to improvements in scientific research methods, enhanced data collection, and refined theories. This meta-analysis suggests that the technology of virtual laboratories has evolved over time, and this research field has received increasing attention. Although the analysis reveals the moderation effect of publication year, considering the representativeness of the samples and limitations of the research design, such as potential selection bias and unequal sample sizes, remains important.

The limitations of the evidence included in the review lie in the risk of bias. First, most of the literature has not provided detailed information about the randomization process or allocation-concealment methods. Some studies have used quasi-experimental designs where the experimental and control groups were based on existing classes rather than random selection, which could have led to selection bias. In some studies, participants were randomly assigned to different experimental conditions, but since they came from specific courses, their backgrounds and abilities might have been uneven, potentially resulting in selection bias. Second, most of the literature has not mentioned whether blinding was applied to the experimental conditions. Although some studies have noted that the experiments were conducted in two different classrooms, which reduced interference between groups to some extent, the involvement of VR and different teaching methods in the experiments might have made teachers and students aware of the experiment's purpose and group assignments, possibly leading to implementation bias. Additionally, some experiments had an imbalanced gender ratio among participants (more men than women or vice versa), which could have affected the generalizability of the results. However, most studies and experiments used standardized measurement tools and statistical analysis methods (such as t-tests, Levene's test, and analysis of covariance [ANCOVA]), which reduced the possibility of other biases to some extent. In conclusion, such studies have not generally provided information on the randomization process or allocation-concealment methods and have not implemented blinding. We hope that future research will be more rigorous in design and implementation to provide more reliable evidence.

The limitations of the review process used in this study include the following aspects: differences in judgment among researchers, which affect the reliability of the literature selection. The participants involved in the literature review and research come from different professional backgrounds. While introducing perspectives from various disciplines can provide a comprehensive evaluation and interpretation of research data, and discussions and collaboration among professionals from different fields can help identify and reduce potential biases in the study, the evaluation of research quality may lack standardization, potentially leading to assessment bias. Additionally, the literature selected for this meta-analysis was limited to English, which might have resulted in the omission of important research findings in other languages.

## 6. Conclusion

This research is a meta-analysis study on the effects of virtual laboratories on engineering education. As an important aspect of STEM education, engineering education is influenced by various factors, and virtual laboratories, as a combination of technological advancements, changes in traditional teaching methods, and shifts in educational policies, have had a forward-looking and revolutionary impact on engineering education. The results of this meta-analysis study indicate that the use of virtual laboratories in engineering education can significantly enhance teaching effectiveness. Therefore, the continued development and utilization of virtual laboratories are essential, with a need to ensure that their future applications align

with the demands of engineering and STEM education fields, studying how to maximize their effectiveness in improving student performance and teaching outcomes.

## Supporting information

**S1 Appendix. All data extracted from the publications that meet the inclusion criteria.**
(XLSX)

**S2 Appendix. Publications identified in the literature search.**
(XLSX)

**S3 Appendix. Statistics for each study.**
(DOCX)

## Acknowledgments

We would like to thank Editage (www.editage.com) for English language editing.

## Author Contributions

**Conceptualization:** Jiaxing Li.

**Data curation:** Jiaxing Li.

**Formal analysis:** Jiaxing Li, Wenhong Liang.

**Funding acquisition:** Wenhong Liang.

**Investigation:** Wenhong Liang.

**Methodology:** Jiaxing Li.

**Resources:** Wenhong Liang.

**Software:** Jiaxing Li.

**Supervision:** Wenhong Liang.

**Writing – original draft:** Jiaxing Li.

**Writing – review & editing:** Jiaxing Li, Wenhong Liang.

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
