## [Decision Letter · Decision Letter 0]

7 Oct 2024

PONE-D-24-28845Effectiveness of Virtual Laboratory in Engineering Education: A Meta-AnalysisPLOS ONE

Dear Dr. Li,

Thank you for submitting your manuscript to PLOS ONE. After careful consideration, we feel that it has merit but does not fully meet PLOS ONE’s publication criteria as it currently stands. Therefore, we invite you to submit a revised version of the manuscript that addresses the points raised during the review process.

We look forward to receiving your revised manuscript.

Kind regards,

Rea Lavi

Academic Editor

PLOS ONE

“Major science and technology projects of Autonomous Region:

Integration and demonstration of key technologies for improving quality, increasing yield and processing comprehensive utilization of walnut

Item No. : (2021A02002-2)”

5. As required by our policy on Data Availability, please ensure your manuscript or supplementary information includes the following:

Additional Editor Comments:

Thank you for submitting your manuscript.

Please revise your manuscript based on the reviewers' comments.

In particular, more work is required to explain your analysis methodology. Thank you.

Reviewers' comments:

Reviewer's Responses to Questions

**Comments to the Author**

1. Is the manuscript technically sound, and do the data support the conclusions?

Reviewer #1: Yes

Reviewer #2: Yes

2. Has the statistical analysis been performed appropriately and rigorously? 

Reviewer #1: Yes

Reviewer #2: Yes

3. Have the authors made all data underlying the findings in their manuscript fully available?

Reviewer #1: Yes

Reviewer #2: No

4. Is the manuscript presented in an intelligible fashion and written in standard English?

Reviewer #1: Yes

Reviewer #2: Yes

5. Review Comments to the Author

Reviewer #1: Review for Effectiveness of Virtual Laboratory in Engineering Education: A Meta-Analysis

The authors have put together a meta-analysis of published research on the impacts and efficacy using simulated lab experiments as compared to physical experiments in the context of engineering education. This is a useful addition to the literature and worth exploring. Overall, the methods and findings are clear and understandable. There are few technical questions that should be addressed and a couple minor style and grammar issues.

Technical:

1. The authors’ goal of summarizing findings into the types of outcomes (academic achievement, skills, etc.) is very interesting. It seems to me that individual experiments may be very different from each other even when addressing the same type of outcome. I assume in these studies there are difference in data type such as quantitative versus qualitative, differences in assessment methods, and differences in protocol. Is the method of Hedge’s g able to help make statements about the effectiveness when the underlying studies are different? If so, I think this argument could be made stronger.

2. I did not understand the objective or reasoning for the analysis of moderator effect. Why where the categories of paper/author nation and year chosen? What was the intended goal of looking at these and what do they show? It seems that there is a fairly small number of studies from the different countries. Does that affect any meaning the reader is to understand from the moderator affect analysis results?

Some minor issues:

• Starting around the methods section, there are a few minor grammatical errors that don’t significantly affect the understanding but can be somewhat confusing or distracting.

• A few of the citations in the figures have formatting error question marks.

• In figure 4, the “L” classification is in two groups. Is there a reason for the separation? If so, that wasn’t conveyed in the paper.

Reviewer #2: You have carried out a large piece of work so well done for the same. It takes a lot of time and energy to carry out SLRs and meta-analysis even more so.

Your work has focussed on an area that is of interest to many engineering educators as laboratories are not always around for everything and also simulation based design is sometimes a great way to develop engineering systems rapidly and in a cost effective way.

The paper could be enhanced by following the following revisions and based on some of the revisions, you will need to demonstrate why this study is actually needed and has not been done in part before (and if so, take that into account and build the rationale for doing this study).

The areas that need improving are:

1) Use a framework (like PICO or other suitable) to frame your research questions, search terms, inclusion/exclusion criteria from the stated objectives as that will help in later organisation of the work. This will be particularly helpful in check list item 5 (how studies were grouped). Also for 10a and 10b in the checklist, this framework approach will help. you need to detail the same however.

2) Need to specify how the quality of the work shortlisted was assessed and how the work of shortlisting and quality assessment was shared with the authors (state clearly who did what and how did you maintain reliability where more than one author did the work). Interrater stats needed as multiple authors are implied to have worked here.

3) For 13a in the checklist - Tabulate each study showing its chars. And comment on how they were eligible for grouping and synthesis. Group studies in a separate table showing how your grouped them for synthesis. Show the full table for 13c.

4) what software was used to calculate statistics. There are standard software available, please use them to make it standard.

5) for 13f, Describe this better than its been done. Process is stated, no evaluation of the values has been presented.

6) for 17 and 18 on the checklist, you must provide this.

7) The discussion section (23 in checklist) is weak and needs more work.

The past SLRs in this areas must be reviewed in a new section thereby carving out the need for this work and what this adds to the body of lit. that other reviews dont already add.

6. PLOS authors have the option to publish the peer review history of their article (what does this mean?). If published, this will include your full peer review and any attached files.

Reviewer #1: No

Reviewer #2: No

---

## [Author Response · Author response to Decision Letter 0]

29 Nov 2024

Responds to each point raised by the academic editor：

Respond: All figures, tables, and appendix have been named as required.

“Major science and technology projects of Autonomous Region:

Integration and demonstration of key technologies for improving quality, increasing yield and processing comprehensive utilization of walnut

Item No. : (2021A02002-2)”

Respond: It has been stated as requested in the cover letter.

Respond: Agree to the data sharing plan.

Respond: A separate title has been added to each figure.

5. As required by our policy on Data Availability, please ensure your manuscript or supplementary information includes the following: 

Respond: It has been added in the attachment named “Appendix 2. Publications identified in the literature search”. 

Respond: It has been added in the attachment named “Appendix 1. All data extracted from the publications that meet the inclusion criteria”.

Respond: It has been added in the attachment named “Appendix 1. All data extracted from the publications that meet the inclusion criteria”.

Respond: There is no missing data.

Respond: No retracted papers were cited. Changes to the reference list have been highlighted.

Responds to each point raised by Reviewer #1：

1. The authors’ goal of summarizing findings into the types of outcomes (academic achievement, skills, etc.) is very interesting. It seems to me that individual experiments may be very different from each other even when addressing the same type of outcome. I assume in these studies there are difference in data type such as quantitative versus qualitative, differences in assessment methods, and differences in protocol. Is the method of Hedge’s g able to help make statements about the effectiveness when the underlying studies are different? If so, I think this argument could be made stronger.

Respond: Thanks for the valuable reviews. As you pointed out, even when addressing the same type of outcomes, there can be significant differences between individual experiments. Hedge's g is a common method used to measure effect size, especially when dealing with mean comparisons (such as differences between two groups). In this study, publications that are irrelevant to the theme and those based solely on students' subjective feedback (such as studies assessing the perceived usability and user satisfaction of various forms of virtual laboratories) have been excluded during the literature screening. Only studies that are relevant to the topic and based on objective assessment criteria (such as skill tests and examinations) and that employed a two-group research design were included. These efforts were made to ensure consistency in research outcomes and types of experiments. Additionally, the results from the sensitivity analysis in section Sensitivity analysis also indicated that the diversity of the studies did not affect the robustness of the results. 

In response to the reviewers' comments, additional clarification has been provided in Section Abstract reading and full text sifting of the manuscript to ensure completeness and rigor in the discussion: publications that were irrelevant to the theme and those based solely on students’ subjective feedback (such as studies assessing the perceived usability and user satisfaction of various forms of virtual laboratories) were excluded. Ultimately, publications that aligned with the theme and were based on objective evaluation criteria (such as skill tests and exams) were included. These studies employed a two-group research design to compare the effects of various forms of virtual laboratories with traditional laboratories on engineering students. Additionally, these studies reported necessary descriptive summary statistics.

 In addition, add the following explanation in section Sensitivity analysis: The overall effect size estimate (Hedges’ g) varies between approximately 0.590 and 0.789 when different studies are excluded, and the change magnitude is approximately 0.2. Thus, overall, the intervention has a positive and significant impact on the outcomes, and the change in effect size is relatively mild, with no significant fluctuations. That is to say, despite the methodological differences among various studies, the results demonstrate good robustness.

2. I did not understand the objective or reasoning for the analysis of moderator effect. Why where the categories of paper/author nation and year chosen? What was the intended goal of looking at these and what do they show? It seems that there is a fairly small number of studies from the different countries. Does that affect any meaning the reader is to understand from the moderator affect analysis results?

Respond: Thank you for raising this question. The reason for selecting country/region and publication year as variables for the moderating effect is to verify whether the context of different countries or regions influences the research results. Additionally, changes in the social environment over time and the continuous development of virtual experimental technologies may impact the research findings. 

Reviewer’s suggestion is very insightful and highlights a flaw in my discussion. To ensure a more comprehensive discussion, additional clarification has been provided in Discussion section: Based on the analysis of moderator effect sizes, we have found that there are certain differences in effect sizes among different countries/regions and publication years. Publications from Nigeria show a significant effect size (2.995), while publications from the United States indicate an extremely low effect size (0.006). This suggests that there may be stronger factors or contextual conditions in Nigeria that make the relevant effects more pronounced, therefore, we speculate that the differences in effect sizes in our research perhaps due to the varying developments in social, economic, or cultural contexts across different countries. However, due to the limited number of studies from different countries, the analysis of the moderating effects in this research can only serve as a hypothesis and reference. We hope that future research will provide more reliable evidence on this issue.

Besides, the effect sizes based on publication year indicate that the recent study (2023) shows a higher effect size (2.340), while the study from 2004 displays a negative effect size (-0.311). This phenomenon often suggests that the latest research provides stronger support and evidence for the phenomenon of interest, which may be related to improvements in scientific research methods, enhanced data collection, and refined theories. In this meta-analysis, it is possible to infer that the technology of virtual laboratories has evolved over time, and there has been an increased attention to this research field. Although the analysis reveals the moderation effect of publication year, it remains important to consider the representativeness of the samples and the limitations of research design, such as potential selection bias and unequal sample sizes. 

Some minor issues:

• Starting around the methods section, there are a few minor grammatical errors that don’t significantly affect the understanding but can be somewhat confusing or distracting.

• A few of the citations in the figures have formatting error question marks.

• In figure 4, the “L” classification is in two groups. Is there a reason for the separation? If so, that wasn’t conveyed in the paper.

Respond: To ensure the accuracy of language use, spelling, and grammar, we utilized the scientific editing services recommended by the journal, Editage, and have explicitly acknowledged this in the acknowledgments section. To distinguish the changes made to the content in this revision, the "Revised Manuscript with Track Changes" document highlights in yellow the modifications made by the authors in response to the reviewers' comments, and highlights in cyan the changes made to the language.

The question raised by the reviewer regarding the classification of "L" in Figure 4 into two groups is due to the involvement of multiple independent studies in some literature, necessitating a distinction in the forest plot. To address this issue, additional clarification has been included in Section Computation of summary effect sizes: It is worth noting that, due to the involvement of multiple independent studies in some publications, the following distinctions have been made in all the forest plots: a suffix of "A" indicates independent studies focused on students' Academic achievement; "O" indicates independent studies focused on students' Operational skill; "C" indicates independent studies on Cognitive ability; "K" indicates independent studies focused on students' Knowledge acquisition; "CL" indicates independent studies on students' Cognitive load; "L" indicates independent studies focused on students' Learning motivation; and "E" indicates independent studies focused on students' Learning Engagement. If the same publication involves multiple similar studies, add Arabic numerals after the uppercase letter suffix for differentiation.

Responds to each point raised by Reviewer #2: 

The paper could be enhanced by following the following revisions and based on some of the revisions, you will need to demonstrate why this study is actually needed and has not been done in part before (and if so, take that into account and build the rationale for doing this study).

1) Use a framework (like PICO or other suitable) to frame your research questions, search terms, inclusion/exclusion criteria from the stated objectives as that will help in later organisation of the work. This will be particularly helpful in check list item 5 (how studies were grouped). Also for 10a and 10b in the checklist, this framework approach will help. you need to detail the same however.

Respond: I appreciate the reviewer’s constructive feedback; it has been very helpful to me. Based on the review comments, in section Define inclusion and exclusion, the application of the PICO framework is explained for constructing the research questions, search terms, and the inclusion/exclusion criteria derived from the established objectives of this study.

2) Need to specify how the quality of the work shortlisted was assessed and how the work of shortlisting and quality assessment was shared with the authors (state clearly who did what and how did you maintain reliability where more than one author did the work). Interrater stats needed as multiple authors are implied to have worked here.

Respond: Based on the review comments, in section Abstract reading and full text sifting, we describe the division of work, reliability maintenance, and feedback on results used in the literature inclusion/exclusion review.

3) For 13a in the checklist - Tabulate each study showing its chars. And comment on how they were eligible for grouping and synthesis. Group studies in a separate table showing how your grouped them for synthesis. Show the full table for 13c.

Respond: It has been added in the attachment named “Appendix 2. Publications identified in the literature search” and “Appendix 1. All data extracted from the publications that meet the inclusion criteria”. The "Appendix 1. All data extracted from the publications that meet the inclusion criteria" lists the intervention characteristics of each individual research, the types of outcomes used as subgroup classification criteria, the reasons for confirming that the individual study meets the screening criteria, and the risk of bias analysis. 

4) what software was used to calculate statistics. There are standard software available, please use them to make it standard.

Respond: Based on the review comments, we have added a description of the literature management software EndNote X8 used to ensure the reliability of information filtering and statistical results in section Abstract reading and full text sifting. And we added the description that we used Comprehensive Meta Analysis software for data calculations and figure creation to ensure the reliability of the results in section Results.

5) for 13f, Describe this better than its been done. Process is stated, no evaluation of the values has been presented.

Respond: The evaluation of the values has been added in section Sensitivity analysis: Sensitivity analysis was performed to evaluate whether the pooled effect size was influenced by individual studies, that is, to assess the influence of

---

## [Editor Report · Decision Letter 1]

9 Dec 2024

Effectiveness of Virtual Laboratory in Engineering Education: A Meta-Analysis

PONE-D-24-28845R1

Dear Dr. Li,

We’re pleased to inform you that your manuscript has been judged scientifically suitable for publication and will be formally accepted for publication once it meets all outstanding technical requirements.

Kind regards,

Rea Lavi

Academic Editor

PLOS ONE

Additional Editor Comments:

Please make sure Tables 1-3 are all easily readable, and also match in style and text size.

---

## [Editor Report · Acceptance letter]

16 Dec 2024

PONE-D-24-28845R1 

PLOS ONE

Dear Dr. Li, 

I'm pleased to inform you that your manuscript has been deemed suitable for publication in PLOS ONE. Congratulations! Your manuscript is now being handed over to our production team.

Kind regards, 

on behalf of

Dr. Rea Lavi 

Academic Editor

PLOS ONE